# Evaluation of Interferon-Gamma Polymorphisms as a Risk factor in Feline Infectious Peritonitis Development in Non-Pedigree Cats—a Large Cohort Study

**DOI:** 10.3390/pathogens9070535

**Published:** 2020-07-03

**Authors:** Emi N. Barker, Philippa Lait, Lorenzo Ressel, Emily-Jayne Blackwell, Séverine Tasker, Helen Kedward-Dixon, Anja Kipar, Christopher R. Helps

**Affiliations:** 1Langford Vets, University of Bristol, Langford BS40 5DU, UK; pip.lait@bristol.ac.uk (P.L.); c.r.helps@bristol.ac.uk (C.R.H.); 2Bristol Veterinary School, University of Bristol, Langford BS40 5DU. UK; emily.blackwell@bristol.ac.uk (E.-J.B.); s.tasker@bristol.ac.uk (S.T.); 3Department of Veterinary Pathology and Public Health, Institute of Veterinary Science, University of Liverpool, Neston CH64 7TE, UK; L.Ressel@liverpool.ac.uk; 4The Linnaeus Group, Shirley, Solihull B90 1BN, UK; 5Circa Healthcare, 116 Dundas Street, Edinburgh EH3 5DQ, UK; helen.kedward-dixon@circahealthcare.com; 6Institute of Veterinary Pathology, Vetsuisse Faculty, University of Zurich, CH-8057 Zurich, Switzerland; anja.kipar@uzh.ch; 7Institute of Global Health, University of Liverpool, Liverpool L1 8JX, UK

**Keywords:** cohort study, feline coronavirus, gamma interferon, genetic risk factor, pyrosequencing

## Abstract

Feline infectious peritonitis (FIP) is a common infectious cause of death in cats, with heritable host factors associated with altered risk of disease. To assess the role of feline interferon-gamma gene (*fIFNG*) variants in this risk, the allele frequencies of two single nucleotide polymorphisms (SNPs) (g.401 and g.408) were determined for non-pedigree cats either with confirmed FIP (*n* = 59) or from the general population (cats enrolled in a large lifetime longitudinal study; *n* = 264). DNA was extracted from buccal swabs or tissue samples. A pyrosequencing assay to characterize the *fIFNG* SNPs was designed, optimized and subsequently performed on all samples. Genotype and allele frequency were calculated for each population. Characterization of the target SNPs was possible for 56 of the cats with FIP and 263 of the cats from the general population. The SNPs were in complete linkage disequilibrium with each other. There was an association between FIP status and genotype (*χ*^2^; *p* = 0.028), with a reduced risk of developing FIP (*χ*^2^; *p* = 0.0077) associated with the genotype TT at both positions. These results indicate that, although *fIFNG* variants may be associated with altered risk of disease, the prevalence of individual variants within both populations limits application of their characterization to breeding purposes.

## 1. Introduction

Feline infectious peritonitis (FIP) is a common infectious cause of death in cats [1]. The causative agent is feline coronavirus (FCoV), which is endemic in the feline population worldwide and commonly only elicits mild or inapparent intestinal disease [2]. Seroprevalence studies have shown that in some multi-cat households over 90% of cats have evidence of exposure to FCoV [3], but that less than 5% of infected cats will go on to develop FIP [4]. 

In addition to viral factors, the increased incidence of FIP within related groups of cats supports the existence of heritable host factors [5,6]. North American and Australian studies have shown pedigree cats, in general, to be at an increased risk of developing FIP, as compared to non-pedigree cats [7,8,9,10], whilst intercontinental variation in which breeds appear to be at greater, or lesser, risk was also evident. In contrast, no association between pedigree status and risk of FIP was detected in a recent German study [1]; however, the risk of individual breeds was not assessed. Numerous studies have implicated the inflammatory cytokine interferon-gamma in the pathogenesis of FIP [11,12,13]. Sequencing of fragments of the feline interferon-gamma gene (*fIFNG*) in a mixed population of cats, with and without FIP, found an increased risk of FIP associated with a heterozygous genotype (CT) at two closely sited SNPs, g.401 and g.408, which were also found to be in complete linkage disequilibrium [14]. Another study identified an increased frequency of these *fIFNG* single nucleotide polymorphisms (SNPs) in pedigree cats with FIP; however, small numbers limited statistical analysis [15]. Based on these very limited data, commercial assays for these and other SNPs are available, with the suggestion that they could be used to indicate genetic risk of FIP. Out-crossing with non-pedigree cats has been suggested to increase genetic diversity and reduce risk of various genetic diseases [16]; however, the prevalence of allele frequency in the general non-pedigree cat population could have a potentially detrimental impact were it unknown.

The aims of the present study is to (i) determine the *fIFNG* allele frequency within a population of non-pedigree cats confirmed as having FIP by histopathology and immunohistochemistry for FCoV antigen; (ii) determine the *fIFNG* allele frequency within the “general non-pedigree cat population” as represented by a large cohort of prospectively-sampled cats recruited into a lifetime longitudinal study for which epidemiological data are available; iii) determine the relative risk conferred by specific *fIFNG* polymorphisms in the development of FIP. It was hypothesized that non-pedigree cats with FIP were more likely to have the heterozygous genotype previously associated with increased risk of FIP.

## 2. Results

### 2.1. Population

All cats from the FIP group had the diagnosis confirmed by immunohistochemistry for FCoV antigen in one or more tissue (*n* = 34) or in effusion pellets (*n* = 25); viral antigen was found to be expressed in lesional macrophages. One duplicate cat was excluded from the effusion pellet group. Where sex was recorded (*n* = 43), 26% were female (*n* = 11) and 74% were male (*n* = 32). Where age was reported (*n* = 47), median age at diagnosis of FIP was 12 months (range 2 to 168 months).

Of the General Population group (*n* = 264), 205 remained in the study (i.e., assumed to be alive), nine had been lost to follow-up (i.e., cat rehomed or owner withdrawn from study) and 50 were deceased as of 1 April, 2020. Of the alive cats, all were >6 years of age (median age 104 months; range 72 to 120 months). Of the cats that were deceased, cause of death was reported by owners in 44 (88%): 24 from road traffic injuries; five from chronic kidney disease; three from neoplasia; two from heart disease; two from neurological problems; two from trauma (unrelated to road traffic injuries); one each of pyothorax, anesthetic complications at neutering, behavioral issues, feline dysautonomia, ”viral infection” and toxoplasmosis. None of the reported clinical signs or diagnoses were indicative of FIP. Where sex was recorded (*n* = 264), 44% were female (*n* = 116) and 56% were male (*n* = 148).

### 2.2. fIFNG SNPs

Characterization of the *fIFNG* g.401 and g.408 SNPs was possible for all tissue samples (*n* = 34), for 92% of the effusion pellets (*n* = 22 of 24) and 99.6% (*n* = 263 of 264) of the buccal swab samples. Table 1 shows the frequency of each genotype and allele at these SNPs for the FIP group and General Population group.

A chi-square test of independence showed that there was no association between sex and genotype for either the FIP group (*χ*^2^ (2, *n* = 41) = 2.61, *p* = 0.271) or the General Population (*χ*^2^ (2, *n* = 263) = 3.19, *p* = 0.203).

A chi-square test of independence showed that there was a significant association between FIP status and genotype, *χ*^2^ (2, *n* = 319) = 7.13, *p* = 0.028. This significance was due to the decreased proportion of TT homozygotes in the FIP group (*p* = 0.0077), whereas there was no significant difference between the proportion of CC homozygotes (*p* = 0.265) and CT heterozygotes (*p* = 0.125). A chi-square test of independence also showed that there was a significant association between FIP status and allele frequency, *χ*^2^ (1, *n* = 638) = 5.83, *p* = 0.016. This significance was due to a decreased frequency of T allele within the FIP group.

## 3. Discussion

This study assessed the prevalence of *fIFNG* SNPs, previously associated with FIP, within non-pedigree domestic cat populations, either from cats previously recruited into a biobank primarily for the study of feline disease and with confirmed FIP, or from cats recruited into a longitudinal cohort study of cat health (representative of the general population). Non-pedigree cats were selected, in part for their genetic heterogeneity (i.e., to minimize breed bias), as they represent over 80% of the cats with FIP [1]. The FIP group statistics were consistent with previous descriptions of cats with FIP i.e., male cats were over-represented, as were young adult cats, albeit with a wide age range [1,17]. As expected, given the presence of *fIFNG* on an autosomal chromosome, there was no association between sex and genotype for the General Population group. The lack of association between sex and *fIFNG* genotype for the FIP group is not supportive of these SNPs influencing the sex bias in the development of FIP.

Nucleotide variations within the genome (e.g., SNPs) may influence activity of the associated protein(s) by altering the translated amino-acid sequence (e.g., by missense or nonsense changes to the exonic sequence, or changes to the intronic sequence resulting in altered splicing of the transcribed messenger RNA) or by relatively increasing or decreasing the amount of the gene that is transcribed and ultimately translated into interferon-gamma (IFN-γ). Sequencing of various genes encoding inflammatory mediators have identified SNPs and their genotype variants with either increased or decreased prevalence in populations of cats with FIP, as compared to control populations [14,18], including SNPs in non-coding intronic regions of *fIFNG* [14]. This led to the suggestion that characterization of feline inflammatory mediator SNPs could be used to guide breeding of cats with increased resistance to FIP [18]. Although our data are consistent with the previous finding of *fIFNG* loci g.401 and g.408 being in complete linkage disequilibrium and an association between these loci and risk of FIP, that is where the agreement ends [14,15]. The initial *fIFNG* study had found cats with FIP (mixed pedigree and non-pedigree) to be twice as likely as the control cats to have heterozygous genotypes at positions g.401/g.408 (62.1%, n = 18/29 cf. 31.7%, n = 26/82; Fisher exact 0.004) [14]. A more recent study of cats (various pedigree breeds) found a similar increased frequency of the heterozygous genotype (59%, n = 13/22 cf. 23%, n = 3/13); however, this was not found to be statistically significant [15]. It should be noted that in both of these studies [14,15], a significant proportion of control (i.e., non-FIP) cats had the heterozygous genotype, whilst ~40% of cats with FIP did not. Although the current study found an association between genotypes at these loci and risk of FIP, this was attributed to the decreased prevalence of TT homozygotes within the FIP group, as compared to the General Population group.

Following initial infection of enterocytes, FCoV infects macrophages and monocytes where it replicates; the latter mediate its systemic spread [19]. In cats that go on to develop FIP, an excessive and inappropriate immune-response results in the monocyte-mediated granulomatous vasculitis and tissue granulomas, sequelae of which include the body cavity effusions and mass lesions [2]. Higher rates of *fIFNG* transcription and IFN-γ concentrations have been found in the blood of healthy cats infected with FCoV as compared to those that have developed FIP, leading to the suggestion that a reduced risk of FIP is associated with a strong cell-mediated (i.e., Th1) immunity [11,12]. Support for dysregulation of the IFN-γ/tumor necrosis factor-alpha (TNF-α) response being a host-associated risk factor in the development of FIP comes from a vaccination study where an increased ratio of IFN-γ to TNF-α was associated with decreased risk of developing FIP, and vice versa [13]. Paradoxically, high concentrations of IFN-γ have been measured in FIP-associated ascitic fluid, despite these cats having low blood concentrations, suggesting that it might not be as simple as an absence of cell-mediated immunity in FIP pathogenesis [12]. Furthermore, transcriptomics have demonstrated inflammatory pathway activation in the mesenteric lymph nodes of cats with FIP, with upregulated transcription of inflammatory cytokines (including IFN-γ) and chemokines [20]. Whether the intronic *fIFNG* SNPs associated with risk of FIP assessed in this study alter IFN-γ production in response to infection with FCoV is unknown, but warrants further investigation. Such studies may be confounded by the suspected polygenic nature of risk of FIP development, or the presence of unknown, but linked, genetic variants [6,18].

Differences in allele/genotype prevalence between this and other studies could be due to the different genetic backgrounds of the populations studied. This is why the two populations used in this study were selected for comparison, so as to minimize collection bias and breed-associated allele frequency bias; however, the possibility for genetic variation between the two populations compared cannot be excluded. This was also a limitation in the previous studies evaluating inflammatory mediators as risk factors in FIP [14,18], if not more so as a combination of both pedigree and non-pedigree cats were included in those populations. It is also possible that were more cats with FIP included then minor risk factors might have been detected; however, these would not have been considered clinically relevant. The optimum control population against which cats with FIP would ideally be compared is subject to debate. The General Population group used for comparison with the FIP group cannot be considered ”FIP-negative” as, although none were reported as having died of FIP, definitive data on cause of death were not available for all deceased cats. Further, a significant proportion of cats in the General Population group remain alive at time of writing and therefore have the potential, albeit very low as they are all now >6 years of age, to go on to develop FIP. However, these limitations are shared by the control groups used in other studies [14,18]. Unlike these previous studies where clinically normal control cats had been or currently were infected with FCoV, whether cats recruited into the General Population group had been or were currently infected with FCoV was unknown. However, as exposure to FCoV should be independent of genetic status, the prevalence of genetic polymorphisms in the general population as a whole would match that of cats from the general population also exposed to FCoV.

## 4. Materials and Methods

### 4.1. Animals and Samples

The FIP group comprised non-pedigree cats of UK origin confirmed as having FIP within the Bristol-Zurich FIP Consortium University of Bristol FIP Biobank or Veterinary Laboratory Services, University of Liverpool archive. The General Population group comprised prospectively sampled non-pedigree cats recruited into a lifetime longitudinal study (Bristol Cats) for which epidemiological data are available. Non-pedigree cats were those recorded as being domestic short-haired, domestic medium-haired or domestic long-haired.

For those in Bristol FIP Biobank, tissue samples were collected at post-mortem examination as previously described [17]. For the cats from the Veterinary Laboratory Services, University of Liverpool, sections of paraffin blocks prepared from formalin-fixed effusion cell pellets were available. These samples had been submitted for immunohistopathology for FCoV antigen for confirmation of effusive FIP [21]. Only basic data, i.e., age, sex and year submitted, were available for most cats. Duplicate cats were excluded.

The Bristol Cats study is a pioneering study of cat health, welfare and behavior set up in 2010 and run by vets, behaviorists and epidemiologists at the University of Bristol [22]. Pet cats of all breeds were recruited into Bristol Cats from 2010 to 2013, and provided buccal swab samples from Spring 2012 to Spring 2015. Owners also completed periodic questionnaires regarding their cat’s health and behavior. The Bristol Cats study database was reviewed for non-pedigree cats for which buccal samples were available and for which permissions to use their data were available (n = 264). Cats were considered to be alive if owners had completed the most recent questionnaire and not requested withdrawal from the study. Cats were considered to be lost to follow-up if owners had requested withdrawal from the study but not reported death. Where reported by owners, cause of death and exact age at time of death was recorded. Where exact dates were not available, the half-way date between the last completed questionnaire and owner report of death or withdrawal from the study was used to calculate age at death or time of censor. Sex was recorded for all cats.

The collection, storage and use of samples used in this project were approved under ethical review by the University of Bristol Animal Welfare and Ethical Review Board (VIN/14/013; VIN/16/020; VIN/18/007; UIN/13/026), whilst the use of samples from the Veterinary Laboratory Services, University of Liverpool archive also falls under the generic approval of retrospective analysis of formalin-fixed paraffin-embedded (FFPE) blocks of animal tissues in the Section of Pathology, Department of Veterinary Pathology and Public Health, University of Liverpool (RETH000942).

### 4.2. DNA Extraction, Amplification and Sequencing

Feline genomic DNA was extracted from samples from the Bristol FIP Biobank as previously described for total nucleic acids [20]. Feline genomic DNA was extracted from sections from the paraffin blocks and from the buccal swabs using a Chemagic 360 automated platform (Perkin-Elmer) in combination with the Chemagic body fluids nucleic acid kit (Perkin-Elmer) and eluted in elution buffer (100 µL).

A PCR to amplify a fragment of the *fIFNG* gene containing the target SNPs was performed using 2x GoTaq Master Mix (Promega), 200nM forward and reverse amplification primers (see Table 2) and 5 μL DNA in a total volume of 25 μL. Thermal cycling was performed in a PTC-200 DNA Engine (MJ Research) with the following thermal profile: 95 °C for 2 min followed by 40 cycles of 95 °C for 20 s, 60 °C for 20 s and 72 °C for 20 s. Pyrosequencing primers (Table 2) were designed using a combination of PyroMark assay design software (Qiagen), Primer3 [23] and MFold [24], and were made by Metabion (Metabion International). Biotinylated PCR products were immobilized on streptavidin-coated Sepharose beads (GE Healthcare UK Ltd.), purified and annealed with the sequencing primer. Pyrosequencing was performed using the PyroMark Q96 platform (Qiagen) according to the manufacturer’s instructions with a nucleotide dispensation order of GCTATAGCACTGTG. Pyrosequencing data were evaluated using PyroMark Q96 v2.5 software (Qiagen Inc.).

### 4.3. Data Analysis

Data (comprising: group (FIP or General Population); cat identification number; age (in months) at time of FIP diagnosis, death, loss from follow-up, or at 1 April, 2020, if alive; diagnosis (FIP; alive; lost to follow-up; died, and cause where recorded); *fIFNG* SNP at g.401 and g.408; reason for exclusion (where applicable)) were entered into a database (Microsoft Excel for Mac v16.16; Appendix A).

SNPs were described relative to their genomic position from the “A” of the start codon of *fIFNG*. Comparisons for each genotype and allele frequency for the cats with FIP vs. the general population (Bristol Cats) were analyzed using a chi-square test. A *p*-value of ≤0.05 was considered statistically significant.

## 5. Conclusions

The results of our study do not support the hypothesis that non-pedigree cats with FIP were more likely to have the heterozygous *fIFNG* genotype than non-pedigree cats in the general population. The results did indicate a negative association between the TT genotype at *fIFNG* g.401/g.408 and the development of FIP. However, as the TT genotype at this position was present in 16% (n = 9/56) of cats with FIP and absent in 66% (173/263) of cats in the general population, and as the previously reported associations between genotype and risk at these loci were not detected in this population, the clinical application of characterization of these SNPs, both on an individual risk basis and to guide breeding programs, cannot be recommended at this time.

## Figures and Tables

**Table 1 pathogens-09-00535-t001:** Total number and frequency percentage of each genotype and allele at the feline interferon-γ gene (*fIFNG*) single nucleotide polymorphisms (SNPs) g.401 and g.408 for both feline infectious peritonitis (FIP) group and General Population group of non-pedigree cats for which sequence data were obtained. *P*-value indicates likelihood that a genotype or allele is associated with FIP status. * For individual cats, identical genotypes were determined at both *fIFNG* g.401 and *fIFNG* g.408 loci, indicative of complete linkage disequilibrium.

SNP		FIP Group (%)	General Population (%)	*p*-Value
*fIFNG* g.401/	CC	16 (28.6)	57 (21.7)	
*fIFNG* g.408 *	CT	31 (55.4)	116 (44.1)	0.028
	TT	9 (16.1)	90 (34.2)	
	Allele C	63 (56.3)	230 (43.7)	0.016
	Allele T	49 (43.8)	296 (56.3)	

**Table 2 pathogens-09-00535-t002:** Primers used in PCR amplification and pyrosequencing of the *fIFNG* gene.

Primer Use	Direction	Sequence
Amplification	Sense	5′-TGGGTATAAAGGACAGTGATGTCG-3′
Amplification	Anti-sense	5′-Biotin-TTCTTCATGCTAACCCTGACCTT-3′
Sequencing	Sense	5′-GATAATTTTGTGGTGAGAATC-3′

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
