# Peer review of "Evaluation of Interferon-Gamma Polymorphisms as a Risk Factor in Feline Infectious Peritonitis Development in Non-Pedigree Cats—A Large Cohort Study"

_pathogens, 2020, doi:10.3390/pathogens9070535_

Round 1

Reviewer 1 Report

The paper assess the role of feline interferon-gamma gene 
and the allele frequencies of two single nucleotide polymorphism as heritable risk factors to develop FIP in non-pedigree cats. This is an interesting and well-written paper.

I have only minor suggestions. I hope these are useful to improve the paper.

Title

I suggest to add “in non-pedigree cats” in the title

Abstract

Line 24-25: “….from a 
cohort enrolled in a large lifetime longitudinal study (n=264)” – is this the control population of non-pedigree cats without FIP?
 If yes please state it

Introduction:

Line 63-64: “….(Bristol-Zurich FIP Consortium in collaboration with 
University of Liverpool Diagnostic Laboratory) “ – maybe this should be cited only in the M&M section and not this in the presentation of the aims

Line 67-69: It was hypothesized that 
cats with FIP were more likely to have the heterozygous genotype previously associated with 
increased risk of FIP.”- I would like to read how this hypothesis was addressed by the results of the study in the Conclusion section

Results

Non-pedigree cats enclosed in this study were DSH or DLH? I suggest add this information. In addition I found the description of “Bristol cats” not very clear, for examples I don’t find their breed and this should clear stated.

P value should always written in the same way, i.e. P or p and .05 or 0.05 (see at line 93, in Table, line 189, lines 89-91)

Author Response

Dear Reviewer 1,

Thank you for taking the time to review our article (pathogens-832180) “Evaluation of interferon-gamma polymorphisms as risk factor in feline infectious peritonitis development – a large cohort study”, for your helpful comments and finally for the opportunity to address them. Our responses are highlighted in italics below.

---

The paper assess the role of feline interferon-gamma gene and the allele frequencies of two single nucleotide polymorphism as heritable risk factors to develop FIP in non-pedigree cats. This is an interesting and well-written paper.

I have only minor suggestions. I hope these are useful to improve the paper.

We thank Reviewer 1 for their kind words and we hope that we have been able to address their concerns.

Title

I suggest to add “in non-pedigree cats” in the title

We thank Reviewer 1 for this suggestion. The title of the revised manuscript has been amended accordingly. (Line 4).

Abstract

Line 24-25: “….from a cohort enrolled in a large lifetime longitudinal study (n=264)” – is this the control population of non-pedigree cats without FIP?
 If yes please state it

We thank Reviewer 1 for raising this interesting point. The cohort of cats enrolled in the lifetime longitudinal study were a comparator population; however, we cannot say that they were ‘without-FIP’ as we cannot guarantee that none of the deceased cats had FIP, despite none being reported as having died of FIP by their owners, or the alive cats still having the potential to go on to develop FIP (or having said that the cats that clearly died of non-FIP related causes, such as road-traffic incidences, having had the potential to develop FIP). We still feel that this is a valid population against which the FIP group can be compared – as the FIP group is compared against an un-biased general population. This line now reads:

“from the general population (cats enrolled in a large lifetime longitudinal study; n=264).” (Line 25-26).

There has been further clarification on this point throughout the manuscript

Introduction:

Line 63-64: “….(Bristol-Zurich FIP Consortium in collaboration with 
University of Liverpool Diagnostic Laboratory) “ – maybe this should be cited only in the M&M section and not this in the presentation of the aims

We thank Reviewer 1 for this suggestion. Mention of sample sources has been removed from the introduction.

Line 67-69: It was hypothesized that 
cats with FIP were more likely to have the heterozygous genotype previously associated with 
increased risk of FIP.”- I would like to read how this hypothesis was addressed by the results of the study in the Conclusion section

We thank Reviewer 1 for this suggestion. A line has now been added to the conclusions to reflect this (Lines 241-243).

Results

Non-pedigree cats enclosed in this study were DSH or DLH? I suggest add this information. In addition I found the description of “Bristol cats” not very clear, for examples I don’t find their breed and this should clear stated.

We thank Reviewer 1 for highlighting this lack of clarity. “Bristol Cats” is the name of longitudinal study into which cats were enrolled, and is entirely independent of breed. We have added a line to the Methods section:

 “Non-pedigree cats were those recorded as being domestic short-haired, domestic medium-haired or domestic long-haired.” (Lines 185-186).

We have also confined mention of the Bristol Cats to the Methods section, replacing it elsewhere in the manuscript with ‘General Population group’.

P value should always written in the same way, i.e. P or p and .05 or 0.05 (see at line 93, in Table, line 189, lines 89-91)

We thank Reviewer 1 for highlighting this inconsistency. We have elected to write use lower case italicised ‘p’ and ‘0.0…’ throughout the manuscript, with a similar format for ‘n’ when indicating number; however, we are happy for the editors to change this to journal style if different.

Reviewer 2 Report

This manuscript was an evaluation of interferon-gamma polymorphisms as a risk factor in FIP.

I think the manuscript needs a further supporting data for this hypothesis.

Points

  1. Please provide information about cat samples that used, i.e. gender, age and health record.
  2. Please provide the method to determine IFN-gamma SNP.

Author Response

Dear Reviewer 2,

Thank you for taking the time to review our article (pathogens-832180) “Evaluation of interferon-gamma polymorphisms as risk factor in feline infectious peritonitis development – a large cohort study”, for your helpful comments and finally for the opportunity to address them. Our responses are highlighted in italics below.

---

This manuscript was an evaluation of interferon-gamma polymorphisms as a risk factor in FIP.

I think the manuscript needs a further supporting data for this hypothesis.

Points

  1. Please provide information about cat samples that used, i.e. gender, age and health record.

As indicated in the methods, only limited data were available for many of the cats in this study. This is due to consent and data protection legalities. In the new supplementary data file we have provided as much detail as we could regarding sex, age at death or time of censor (e.g. if lost to follow-up or alive), and whether they are from the FIP group or from the large cohort study. We have also included population statistics (sex, age) for the two populations (FIP vs. General) and compared these to genotype – we had already performed these calculations, but as they had either matched results previously reported in other studies (i.e. FIP most frequently affects younger male cats) or revealed no associations (i.e. between genotype and sex), as had been expected, had not been included. As this and Reviewer 3 feel that the paper would benefit from these data and analyses, these have now been added.

  1. Please provide the method to determine IFN-gamma SNP.

Further information has been added to clarify how the SNP was determined:

“Biotinylated PCR products were immobilised on streptavidin-coated Sepharose beads (GE Healthcare UK Ltd.), purified and annealed with the sequencing primer Pyrosequencing was performed using the PyroMark Q96 platform (Qiagen) according to the manufacturer’s instructions with a nucleotide dispensation order of GCTATAGCACTGTG. Pyrosequencing data were evaluated using PyroMark Q96 v2.5 software (Qiagen Inc.).” (Lines 224-228).

Apologies, we also noted that table 2 (describing the primers used) was also missing from the original manuscript. This has now been added (Line 229).

Reviewer 3 Report

It is very important to understand the critical factors affecting the development of FIP after the infection of FCoV. Feline interferon-gamma polymorphisms were evaluated for risk factor in this study. Bristol FIP Biobank or Veterinary Laboratory Services, University of Liverpool archive provides a valuable resources for this large cohort study. However, no information of breed background for both populations is available for comparison even though the authors mentioned the importance of breed background. In addition, it is unclear whether all participated cats were collected samples for FCoV detection/FIP diagnosis every year or the status of "FIP-negative" was based on one-time swab sample and the periodic questionnaires. I will recommend that more statistic analyses on more factors of genetic backgrounds and FCoV/FIP status. If the analyses have been performed before, further discussion and comparisons would be much appreciated.    

Author Response

Dear Reviewer 3,

Thank you for taking the time to review our article (pathogens-832180) “Evaluation of interferon-gamma polymorphisms as risk factor in feline infectious peritonitis development – a large cohort study”, for your helpful comments and finally for the opportunity to address them. Our responses are highlighted in italics below.

---

It is very important to understand the critical factors affecting the development of FIP after the infection of FCoV. Feline interferon-gamma polymorphisms were evaluated for risk factor in this study. Bristol FIP Biobank or Veterinary Laboratory Services, University of Liverpool archive provides a valuable resources for this large cohort study. However, no information of breed background for both populations is available for comparison even though the authors mentioned the importance of breed background.

There were two groups of cats. Firstly, those definitively diagnosed with FIP from the Bristol FIP Biobank and University of Liverpool archive. Secondly, the larger cohort comprised cats enrolled in the lifetime longitudinal (i.e. the ‘Bristol Cats’) study, independent of health status – to reflect the general non-pedigree population. As indicated, only non-pedigree cats from the UK were included in both groups of cats – to eliminate any breed-associated differences between the two populations. We have added a line to the Methods section: “Non-pedigree cats were those recorded as being domestic short-haired, domestic medium-haired or domestic long-haired.” (Lines 185-186) to help make this clearer to the reader.

In addition, it is unclear whether all participated cats were collected samples for FCoV detection/FIP diagnosis every year or the status of "FIP-negative" was based on one-time swab sample and the periodic questionnaires. I will recommend that more statistic analyses on more factors of genetic backgrounds and FCoV/FIP status. If the analyses have been performed before, further discussion and comparisons would be much appreciated. 

Given that the two groups of cats (i.e. those with FIP, and those from the general population) all comprised non-pedigree ‘domestic’ cats of variable coat length, it is unclear as to which statistical analyses Reviewer 3 is asking for. Cats from the longitudinal cohort were not recruited based upon health status – which is the one of the advantages of this group, as this eliminates bias. The cats in the longitudinal cohort are not described as being FIP-negative, as although none of the owners reported that their cats had been diagnosed with FIP this cannot be definitively excluded as cause of death and was not reported for some cats.

            We presume Reviewer 3 was referring to detection of FCoV rather than diagnosis of FIP from buccal swab samples. The aim of this study was to determine whether those cats that have developed FIP are more likely to have certain fIFNG polymorphisms, as compared to the population as a whole. Whether or not that general population has been exposed to FCoV is not relevant to this, as exposure to FCoV would be independent of genetic status. As the genetic status of individual cats does not change over the course of their lifetime the timing of swab collection is irrelevant, and repeat sample testing unnecessary. The revised manuscript has been amended to hopefully address these issues more clearly for the reader: with additional data and analyses regarding sex added to the results (Lines 91-93) and discussion (Lines 116-118), as well as further discussion as to limitations to various control populations used in this and other studies in relation to FCoV-infection / exposure status (Lines 167-178). We hope that this addresses the concerns of Reviewer 3.

Round 2

Reviewer 2 Report

This paper is improved and well written. However, there seems to be lack of support data to warrant this hypothesis. This will be the next study.
